# Current Worldwide Trends in Pediatric *Helicobacter pylori* Antimicrobial Resistance

**DOI:** 10.3390/children10020403

**Published:** 2023-02-18

**Authors:** Reka Borka Balas, Lorena Elena Meliț, Cristina Oana Mărginean

**Affiliations:** Department of Pediatrics I, “George Emil Palade” University of Medicine, Pharmacy, Sciences and Technology, Târgu Mureș, Gheorghe Marinescu Street, No. 38, 540136 Târgu Mureș, Romania

**Keywords:** *Helicobacter pylori*, antimicrobial resistance, children

## Abstract

*Helicobacter pylori* (*H. pylori*) has acquired several resistance mechanisms in order to escape the currently used eradication regimens such as mutations that impair the replication, recombination, and transcription of DNA; the antibiotics capability to interact with protein synthesis and ribosomal activity; the adequate redox state of bacterial cells; or the penicillin-binding proteins. The aim of this review was to identify the differences in pediatric *H. pylori* antimicrobial-resistance trends between continents and countries of the same continent. In Asian pediatric patients, the greatest antimicrobial resistance was found to metronidazole (>50%), probably due to its wide use for parasitic infections. Aside from the increased resistance to metronidazole, the reports from different Asian countries indicated also high resistance rates to clarithromycin, suggesting that ciprofloxacin-based eradication therapy and bismuth-based quadruple therapy might be optimal choices for the eradication of *H. pylori* in Asian pediatric population. The scarce evidence for America revealed that *H. pylori* strains display an increased resistance to clarithromycin (up to 79.6%), but not all studies agreed on this statement. Pediatric patients from Africa also presented the greatest resistance rate to metronidazole (91%), but the results in terms of amoxicillin remain contradictory. Nevertheless, the lowest resistance rates in most of the African studies were found for quinolones. Among European children, the most frequent antimicrobial resistance was also noticed for metronidazole and clarithromycin (up to 59% and 45%) but with a predominance for clarithromycin as compared to other continents. The differences in antibiotic use among continents and countries worldwide is clearly responsible for the discrepancies regarding *H. pylori* antimicrobial-resistance patterns, emphasizing the crucial role of global judicious antibiotic use in order to control the increasing resistance rates worldwide.

## 1. Introduction

*Helicobacter pylori* (*H. pylori*) infection occurs usually during early childhood, and its prevalence increases with age is probably the most-studied infection worldwide. Nevertheless, if left untreated, it might result not only in life-threatening local complications such as peptic ulcer disease, gastric adenocarcinoma, or mucosa-associated lymphoid tissue lymphoma during adulthood but also systemic complications related to the subclinical inflammation, including iron deficiency anemia, thrombocytopenic purpura, growth delay, vitamin B_12_ deficiency, and even cardiovascular, metabolic complications, or autoimmune diseases such as thyroiditis, which has an increased prevalence in our country [1,2,3]. Wide variations regarding the prevalence of this infection were reported across continents, countries, and even geographical areas of the same country [4]. Still, the studies performed during the last two decades have pointed out a descending trend even in developing countries, most likely due to the improvements in hygiene and sanitary conditions worldwide [4]. Several risk factors were proven to increase the risk of *H. pylori* infection depending on geographical area, such as male gender, ethnicity, rural residence, difficult access and poor adherence to treatment, poverty, poor living conditions, birth order, using a traditional pit or having no toilet, no in-home water service, source of drinking water, drinking water from tanks, farmer profession, educational level, lower parental education and unemployment, lower income, improper sanitation, not washing hands after school, house crowding, eating spicy food and raw uncooked vegetables, eating unwashed vegetables and fruits, active smoking, and alcohol drinking [4]. Prenatal transmission from infected mother or the transmission of this infection during delivery were also hypothesized as potential risk factors incriminated in increasing *H. pylori* prevalence in the pediatric population [5], but these findings remain controversial [6,7]. 

The previously mentioned environmental factors are synergically potentiated by *H. pylori* virulence factors and the host’s innate immune responses, which contribute to the long-term persistence of this bacterium within the gastric mucosa, also triggering its systemic complications. Therefore, *H. pylori* virulence factors include lipopolysaccharides, flagellin, vacuolating cytotoxin A, cag-pathogenicity island, adhesins, and pathogen-associated molecular patters, while innate immune responses involve the action of toll-like receptors, proinflammatory cytokines, chemokines, and chemotactic proteins but also other cells such as monocytes, macrophages, neutrophils, natural killer cells, dendritic cells, or T and B lymphocytes [8,9,10,11,12,13]. A recent study underlined that pediatric patients harboring cagA- and cagE-positive *H. pylori* strains are less susceptible to *H. pylori* eradication treatment [14]. Similar findings were reported also by Karbalaei et al., who emphasized that cagA-positive strains express increased resistance to metronidazole, while those carrying the vacA s1m1 genotype resulted in increased resistance to amoxicillin and levofloxacin [15]. Moreover, Karabiber et al. pointed out that cagE and vacA s1 a/m2 genotypes represent major virulence factors associated with increased antibiotic resistance [16]. Aside from cagA, vacA s1m1, and cagE, other studies proved that iceA2 and babA2 are also associated with *H.-pylori*-positive gastritis severity [17,18]. All these cells create a bridge between local and systemic inflammation in the setting of *H. pylori* infection. 

Diagnosis and proper eradication might break this vicious circle, proving that effective eradication is the cornerstone in lowering the risk of gastric cancer [19,20]. Several important challenges preempt the proper eradication, such as questions regarding which is the most effective testing method, who should be tested, or who should be treated [1].

Bacterial resistance rates were recently reported to present an increasing trend not only in terms of clarithromycin and metronidazole but also when using drugs such as levofloxacin, which is considered a rescue therapy [21]. Guidelines recommend the standard triple therapy consisting of two antibiotics (clarithromycin, amoxicillin, or metronidazole) along with a proton pump inhibitor as eradication regimen for *H. pylori* infection [22,23]. This standard therapy has been used worldwide since 1990s, when it was proven that its eradication rate reaches up to 98% [24]. The current reports have lately indicated a considerable decrease of these rates to less than 70%, raising important concerns regarding the optimal eradication of this bacterium, especially in pediatric patients [25,26]. It was also proven that antimicrobial-resistance rates vary across different geographical areas depending on the local policies for antibiotic use. This statement is sustained by several studies that indicated resistance rates varying from 49% to 1% for clarithromycin [27] and 100% to 17% for metronidazole [28,29,30,31]. Based on these findings, experts have tried to find different alternative regimens to increase the eradication rates in those who fail to respond to the standard regimen including bismuth quadruple therapy (tetracycline, metronidazole, bismuth, and proton pump inhibitor for 14 days); non-bismuth quadruple therapy (clarithromycin, metronidazole, amoxicillin, and proton pump inhibitor for 10–14 days); sequential therapy (proton pump inhibitor + amoxicillin for 7 days, followed by 7 days of quadruple therapy); and the replacement of clarithromycin by levofloxacin in triple or sequential therapies [32]. Nevertheless, a recent review pointed out that in most World Health Organization regions, both primary and secondary *H. pylori* resistance to metronidazole, clarithromycin, and levofloxacin exceed 15%, which represents the common threshold for selecting alternative empiric regimens [33]. Moreover, recent reports mention wide variations regarding the efficacy of the previously mentioned alternative regimens [34,35], and although international consensus recommends choosing the eradication regimen based on local resistance characteristics, this testing is not commonly performed [36,37].

Several mechanisms were found to be involved in *H. pylori* antimicrobial resistance that in fact are common to other bacterial infections in humans, such as mutations that impair the replication, recombination, and transcription of DNA; mutations that affect the antibiotics capability to interact with protein synthesis and ribosomal activity; mutations altering the adequate redox state of bacterial cells, resulting in impaired activity of oxireductases; as well as mutations that change penicillin-binding proteins, contributing to peptidoglycan biosynthesis and the specific target of β-lactams action [38]. Moreover, outer membrane poring and efflux systems were also proven to strongly contribute to drugs tolerance due to their ability to maintain a low concentration of the toxic agent within the bacterial cells, resulting in a decreased capacity of antibiotics to kill bacteria [39]. Certain bacterial peculiarities were also revealed to reduce susceptibility to antibiotics, such as its ability to shift between rod-shaped active bacteria and dormant resting coccoidal state as a response to antibiotics but also the capacity to form biofilms on gastric mucosa surface after its penetration [40]. 

In spite of the current descending trend in *H. pylori* prevalence, we are far from understanding the complexity of its long-term persistence and even further from finding the ideal eradicating model based on the combination of the interaction between *H. pylori* virulence factors, antimicrobial-resistance features, the peculiarities of host’s immune responses, and environmental factors. 

The aim of this review was to assess the trends of pediatric *H. pylori* antimicrobial resistance worldwide in order to guide the eradication treatment according to geographical peculiarities.

## 2. Literature Search

We performed an electronic search of the literature on the lines of search for a narrative overview based on three databases, namely *PubMed*, *EMBASE*, and *Google Scholar*, using the following search terms: “*Helicobacter pylori* antimicrobial resistance”, “children”, as well as names of countries from each continent. *Google Scholar* was used as an option when the full text of the article was not found in the other two databases. We included articles published in the last ten years involving pediatric patients on the topic of *H. pylori* antimicrobial resistance in order to underline the differences in resistance trends worldwide. We critically assessed all included articles depending on the suitability of the methods used to check the hypothesis, the key results, the interpretation of the results, the quality of the results, the limitations of the study, and the relevance of the conclusions to the field [41]. We used the critical reading tools for scientific article proposed by Jean-Baptist du Prel et al. [42] involving the assessment of the design and structure of each article, the relevance of each section, but also the possible sources of limitation and bias. Based on these theories, the most reliable critical reading tools used to assess the articles included in this review were analysis and inference. All articles that did not match these criteria were excluded from the study as well as those written in other languages than English or those that had only the abstract available. When assessing the articles, we also identified supplementary references that matched our criteria, and we performed a manual search in the reference list of each article retrieved in the initial round of search. For these articles, we followed the same method as described above before including them in the review. Eventually, we synthetized the results from all articles included in the study in order to underline both the similarities and discrepancies between them for providing an objective narrative overview regarding the trends of *H. pylori* antimicrobial resistance worldwide.

## 3. Pediatric *H. pylori* Antimicrobial Resistance in Asia

A recent review assessing the worldwide *H. pylori* antibacterial resistance pointed out a primary clarithromycin-resistance rate of 10% in the Southeast Asia region, which is identical to the American region but lower than in Europe and eastern Mediterranean regions (>15%) [33]. In terms of primary resistance to metronidazole, the review indicated a rate of 51% in the same Asian region, while for amoxicillin and tetracycline, the primary resistance rate was of 2% or less. Surprisingly, the authors identified a high primary resistance rate to levofloxacin of 30%. The review also analyzed the secondary resistance rates and underlined the following findings for the Southeast Asia region: a secondary resistance rate for clarithromycin was 15% as compared to 44% for metronidazole and 24% for levofloxacin. No available data were found in this area for secondary resistance rates to amoxicillin, tetracycline, or combined clarithromycin and metronidazole. Moreover, after analyzing the trends, the authors identified that the clarithromycin-resistance rate increased significantly from 13% in 2006–2008 to 21% in 2012–2016 [33] (Table 1).

Studies performed in Turkey indicated high rates of *H. pylori* resistance to clarithromycin and metronidazole, revealing an eradication rate of only 60% or less as a result of the standard triple therapy [43], with a decreasing trend [5]. Moreover, the resistance rates were reported to vary between 24.8% to 36.7% for clarithromycin and 33.7% to 35.5% for metronidazole in the same Asian country [44,45]. For other antibiotics used for *H. pylori* eradication, the prevalence of the Turkish antimicrobial-resistance rates was 23.7% for levofloxacin, 3.5% for tetracycline, and 0.9% for amoxicillin [45]. Further studies highlighted that the resistance rates to clarithromycin reached almost 40% in Turkey [46]. Regarding the eradication regimens, studies on the Turkish population revealed a decrease in sequential therapy eradication rate (to 80%) along with a better eradication rate of up 96.4% for bismuth-containing quadruple therapy as well as up to 95% for levofloxacin-containing therapies [46]. Based on these findings, a recent study including Turkish children suggested that adding bismuth to the sequential therapy and using amoxicillin and tetracycline might be successful options for increasing *H. pylori* eradication rates in this population [47]. Still, clarithromycin-resistance rates were found to vary even among different areas of Turkey since, according to a study from Manisa region performed on children, clarithromycin might still be used as a first-line regimen for eradicating *H. pylori* infection, but genetic mutations such as A2143G and A2142G should be considered in those who do not respond to clarithromycin-based regimens [48]. Moreover, the presence of hetero-resistant strains could also contribute to the high rate of eradication failure in children form Turkey [49] (Table 1).

Similar findings were also reported in children from China, emphasizing that the *A2143G* mutation was the most common in clarithromycin-resistant strains, while *Asn87* along with Asp91 of the gyrA mutation point were the most frequently encountered in the levofloxacin-resistant strains [50]. Moreover, Zhang et al. emphasized that in Chinese children, most mutations appeared at A2143G of the 23S rRNA gene for clarithromycin, 91 mutation of gyrA gene for levofloxacin, and G47A mutation of rdxA gene for metronidazole [51]. Genetic studies were performed also on Chinese children assessing the role of tailored therapy (phenotype-guided therapy) on traditional culture results versus genetic testing results, i.e., genotype-guided therapy [52]. The study proved that the eradication rates of the latter therapy were superior to that of phenotype-guided therapy, reaching up to 92%. Therefore, we emphasize the fact that genotype-guided therapy represents the future in *H. pylori* eradication therapy, and although it is extremely expensive, it should be used in children in whom tailored therapy guided by tradition culture fails in eradicating this bacterium. Another study performed on children form Southwest China including three ethnic groups, i.e., Han, Tibetan, and Yi, found overall resistance rates of 71.3% for metronidazole, 60.9% for rifampicin, 55.2% for clarithromycin, and 18.4% for levofloxacin, with no strains expressing resistance to amoxicillin, tetracycline, or furazolidone [53]. After dividing the sample into treatment-naïve patients and previously treated patients, the authors noticed for the first group the following resistance rates: 73.6% for metronidazole, 60.4% for rifampicin, 45.3% for clarithromycin, and 15.1% for levofloxacin; for the second group, they noticed the highest resistance rate for clarithromycin (70.6%), followed by metronidazole (67.6%), rifampicin (61.8%), and levofloxacin (23.5%). The authors emphasized the use of bismuth quadruple therapy in this population as a first-line regimen instead of clarithromycin-based triple therapy [53]. Similar findings were reported in children from Hangzhou, China, by Shu et al., who found no resistant strains to amoxicillin, furazolidone, and gentamycin but noticed total resistance rates to clarithromycin of 20.6%, 68.8% for metronidazole, and 9% for levofloxacin [54]. Moreover, the authors noticed a significant increase in *H. pylori* antimicrobial resistance from 2012 to 2014. A more complex study from China involving treatment-naïve adults, previously treated adults, treatment-naïve children, and a sample of adults from a population included in a health survey noticed increased resistance rates for clarithromycin, metronidazole, and levofloxacin, especially in the group of previously treated adults, and lower resistance rates to clarithromycin in treatment-naïve patients aged between 10–24 years [55]. Surprisingly, Tang et al. noticed that although the *H. pylori* prevalence decreased in China during the last 12 years, the failure of first-line antibiotic therapy considerably increased [56]. Nevertheless, Tong et al. concluded that 7-day triple therapy regimens remain a viable option in populations with low resistance rates [57] (Table 1).

A review article on *H. pylori* eradication strategies in children and adolescents from Korea concluded that bismuth-based quadruple therapy consisting of bismuth, amoxicillin or tetracycline, metronidazole, and a proton pump inhibitor administered for 14 days with maximal tolerable doses can be used in this Asian country, taking into account the lack of initial testing for *H. pylori* antibiotic susceptibility in this population [58]. Nevertheless, if this regimen also fails in eradicating *H. pylori*, data from Korea showed that rifabutin might represent an option in this population since *H. pylori* strains resistant to rifabutin were found to be rare in both children and adults living in Korea, but close monitoring of this therapy is required since myelosuppression as well as the development of rifamycin-resistant strains of *Mycobacterium tuberculosis* are common in those using rifabutin-containing regimens (Table 1).

An increase in resistant *H. pylori* strains was also noticed in Vietnam [59]. Thus, a recent review in this region of Asia found a primary resistance rate of 69.4% for metronidazole, followed by 48.8% for multidrug resistance, 34.1% for clarithromycin, 27.9% for levofloxacin, 17.9% for tetracycline, and 15% for amoxicillin. In terms of secondary resistance rates, the authors revealed the highest rate for clarithromycin (74.9%), closely followed by multidrug resistance (62.3%) and metronidazole (61.5%), slightly lower for levofloxacin (45.7%), 23.5% for tetracycline, and the lowest for amoxicillin (9.5%) [59]. Another study from Mekong Delta, Vietnam, involving pediatric patients showed the following resistance rates: 80.6% to clarithromycin, 71.7% to amoxicillin, 49.4% to metronidazole, 45.1% to levofloxacin, and 11.4% to tetracycline [60]. Moreover, the overall eradication rate found in this study reached 83.1%, with the greatest success achieved by bismuth quadruple therapy tailored to antimicrobial susceptibility. Thus, the authors concluded that bismuth and tetracycline should be used in this area of Vietnam in order to increase the eradication rates in children [60]. Thieu et al. also aimed to identify *H. pylori* antimicrobial-resistance rates in Vietnamese children and noted an alarming rate for clarithromycin (92.1%), followed by amoxicillin (50%), levofloxacin (31.6%), metronidazole (14.5%), and tetracycline (0%) [61]. In terms of adding bismuth to the standard therapy, the study noticed that it might increase the eradication rate by 3.69-fold as compared to therapies without bismuth. Based on these findings, it is worth mentioning that *H. pylori* resistance trends might differ even among the different areas of the same country, requiring larger studies even within the same country involving populations from different areas in order to obtain a proper characterization of *H. pylori* resistance trends (Table 1).

The results from Israel were found to considerably differ between children that received previous treatment and those who were not previously treated. Kori et al. performed a study on 123 strains of *H. pylori* and noticed that in treatment-naïve children, the primary global resistance was 38%, with a 32.6% resistance rate for metronidazole, 9.5% for clarithromycin, and 4.2% for both [62]. When assessing the resistance rates in previously treated children, the authors noticed that the resistance rate to clarithromycin increased to 29%, for metronidazole to 61%, and for both to 18%. The study found that all the assessed *H. pylori* strains were susceptible to amoxicillin, levofloxacin, and tetracycline [62]. Therefore, the authors emphasized the need to reassess the statement “test and treat” and to use, when possible, the culture-based treatment strategy. Another study from Israel that compared children and adults in terms of *H. pylori* antimicrobial-resistance patterns noticed that amoxicillin resistance rates were higher in adults as compared to children, and increased secondary resistance rates to metronidazole and clarithromycin were high in both pediatric and adult patients, but no levofloxacin-resistant strains were identified in children [63] (Table 1).

Studies from Iran pointed out that *H. pylori* strains in a pediatric population from the northern region present the highest resistance rate to metronidazole, followed by furazolidone, and the lowest to clarithromycin, contradicting almost all the findings reported above in other regions of China [64]. According to a study performed on Iranian children, ciprofloxacin might be an option in eradicating multidrug-resistant *H. pylori* strains [65]. The study divided the sample into two groups: the first group received ciprofloxacin, amoxicillin, and omeprazole, and the second who was administered amoxicillin, furazolidone, and omeprazole, identifying an eradication rate of 87.9% in the first group and 60.6% for the children included in the second group. A more recent review performed on Iranian children indicated the following resistance rates: 71% for metronidazole, 28.6% for rifampicin, 21.4% for ampicillin, 20.4% for amoxicillin, 19% for azithromycin, 16.2% for ciprofloxacin, 15.3% for erythromycin, 12.2% for clarithromycin, and 8.4% for both tetracycline and furazolidone, while for nitrofurantoin, no resistant strains were reported [66] (Table 1).

The increased *H. pylori* resistance rates to metronidazole in Asia might be related to its wide use as an antiparasitic agent in these areas since it is well known that parasitic infections are frequent in the Asian population. Surprisingly, most of the studies in these countries reported relatively high resistance rates to levofloxacin, which might result in reducing the success of eradication regimens based on the fact that levofloxacin is considered a rescue therapy.

## 4. Pediatric *H. pylori* Antimicrobial Resistance in America

Unlike Asia, the reports from America regarding *H. pylori* antimicrobial-resistance rates in children are scarcer. Savoldi et al. highlighted in their recent review a primary clarithromycin resistance of 10% in the Americas region, while the secondary resistance to the same antibiotic increased to 18% [33]. In terms of primary resistance to metronidazole in America, the authors found a rate of 23%, which increased to 30% when assessing secondary resistance rates. The same increasing pattern was found for resistance to levofloxacin: 15% for primary resistance and 22% for secondary resistance. On the other hand, when assessing *H. pylori* resistance to amoxicillin, the authors found a decrease from 10% to 7% between primary and secondary resistance rates [33]. The study found no resistant strains to either tetracycline or clarithromycin combined with metronidazole. 

Additionally, a study performed on children from a rural community of Cajamarca, Peru, revealed clarithromycin-resistant mutations in 79.6% of the patients who tested positive for *H. pylori* infection [67]. The authors found the A2142G mutation together with the double mutations A2142G–A2143G to be the most common mutations in terms of clarithromycin resistance among the studied pediatric population. Similar findings were reported by Miftahussurur et al. in the Dominican Republic regarding the A2142G mutation inducing resistance to clarithromycin, but additionally, the authors pointed out that T1958G was also responsible for clarithromycin resistance [68]. The same study also showed an association between Ser-14 of trx1 and Arg-221 of dapF mutations and metronidazole resistance as well as between the Asn-87 mutation of gyr2 gene and levofloxacin resistance, confirming an increased *H. pylori* resistance pattern to both metronidazole and levofloxacin in children from the Dominican Republic [68]. Ogata et al. proved that Brazilian children and adolescents infected with *H. pylori* also express an increased resistance to metronidazole (40%) and clarithromycin (19.5%), followed by amoxicillin (10.4%) but no resistance to furazolidone and tetracycline [28]. on the other hand, low resistance rates to clarithromycin (8%) were noticed in Colombian children who tested positive for *H. pylori* infection [69]. Nevertheless, a study performed on a pediatric population from Chile, which is known to have an increased risk of gastric cancer, also reported increased resistance rates to clarithromycin, showing that 67% of the patients in whom the clarithromycin-based standard triple therapy failed carried A2143G mutations [70]. 

Although in American children, most of the studies revealed *H. pylori* strains resistant to clarithromycin and metronidazole, several reported indicated the contrary imposing the need for further studies in order to delineate clear recommendations for eradicating *H. pylori* infection in different areas of America (Table 1). The reported high resistance rates to both metronidazole and clarithromycin, as well as the increasing trend regarding levofloxacin resistance rate in this geographic area might be related to their wide use for other bacterial infections probably sometimes unjustified. Nevertheless, studies indicated that if used together, clarithromycin and metronidazole seem to maintain their effectiveness in terms of *H. pylori* eradication. 

## 5. Pediatric *H. pylori* Antimicrobial Resistance in Africa

The reports regarding pediatric *H. pylori* resistance trends in Africa are even fewer as compared to America. According to the review of Savoldi et al., the highest overall resistance rates in this population were found for metronidazole (91%), followed by amoxicillin (38%), clarithromycin (15%), levofloxacin (14%), and tetracycline (13%) [33]. Similarly, Jaka et al. analyzed 26 articles in which the tested *H. pylori* strains showed the greatest resistance to metronidazole (75.8%) and amoxicillin (72.6%) but also tetracycline (48.7%) and clarithromycin (29.2%) [71]. Only 17.4% of the *H. pylori* isolated expressed resistance to quinolones in African children.

On the contrary, a recent study involving Algerian children found no strains resistant to either amoxicillin or combined metronidazole and clarithromycin but revealed an increased resistance rate to metronidazole (37%) and clarithromycin (13%) [72]. In terms of *H. pylori* eradication rate, sequential therapy showed a significantly higher eradication rate as compared to compared to conventional triple therapy in children from Kenya [73]. Moreover, a regimen consisting of amoxicillin and higher doses of metronidazole (40 mg/kg/day) along with a proton pump inhibitor was also proven to be effective in eradicating *H. pylori* infection in Africa even in metronidazole-resistant strains [72]. Shawki et al. studied the efficacy of a novel regimen consisting of nitazoxanide, clarithromycin, and a proton pump inhibitor in children from Egypt and noticed that this therapy carries a lower risk of resistance rates and increases the chance of eradication in comparison to standard triple therapy involving metronidazole, clarithromycin, and a proton pump inhibitor [74]. 

Undoubtedly, similar to the reports from Asia, *H. pylori* resistance to metronidazole is a major problem in children from Africa, most likely due to its wide use in parasitic infections, which are highly prevalent in this population (Table 1). Nevertheless, as compared to Asian and American children, the resistance rates for levofloxacin in the African pediatric population were found to be lower. 

## 6. Pediatric *H. pylori* Antimicrobial Resistance in Europe

Based on the findings of Savoldi et al., the primary resistance rates of *H. pylori* to clarithromycin, metronidazole, levofloxacin, and clarithromycin combined with metronidazole were 18%, 32%, 11%, and 1%, respectively [33]. Concerning secondary resistance rates, the authors reported 48% for both clarithromycin and metronidazole, 19% for levofloxacin, and 18% for the combination between clarithromycin and metronidazole. The review found no strains showing primary or secondary resistance to either amoxicillin or tetracycline. 

A recent study on Spanish children pointed out that more than two-thirds of the children infected with *H. pylori* carried resistance to at least one antibiotic, while 16.3% were found harboring double-resistant strains [75]. Moreover, the study found the following resistance rates: 44.9% for clarithromycin, 16.3% for metronidazole, 7.9% for levofloxacin, and 2% for amoxicillin. Similar resistance rates to clarithromycin and metronidazole were also identified by Montes et al. in pediatric population from Spain: 34.7% for clarithromycin and 16.7% for metronidazole [76]. 

Most of the studies regarding *H. pylori* resistance trends were performed in Poland. Thus, a study published in 2014 noticed that *H. pylori*-infected children living in Poland displayed the greatest resistance rate to metronidazole (44.8%), followed by clarithromycin (33.3%), while in 1.9% of the cases, the strains showed simultaneous resistance to clarithromycin, metronidazole, and levofloxacin [77] (Table 1). A more recent study of the same author published in 2017 revealed a decrease in metronidazole resistance rates (27.8%) and a mild increase in the clarithromycin-resistance rate (38.9%) [78]. In addition, they noticed that almost 50% of the Spanish children included in this study presented significant heteroresistance, with the combined genotype cagA + vacA s1/m2 as the predominant genotype in the detected isolates of *H. pylori* [78]. These findings were sustained by other studies on both Polish children and adults [79,80] (Table 1). Furthermore, studies in Poland emphasized the role of therapy guided by antibiotic resistance in successful eradication of *H. pylori* infection [81]. Therefore, it was emphasized that triple standard therapy consisting of amoxicillin + clarithromycin + a proton pump inhibitor as well as sequential therapy might be the best choice in children with *H. pylori* strains susceptible to clarithromycin [81]. Unfortunately, the resistance rates in Polish population have increased during the past 10 years, according to a study published in 2020, where the authors noticed resistance rates to clarithromycin and metronidazole of 54.5% and 31.8%, respectively [82]. Even more concerning is that the authors showed significantly higher resistance rates to clarithromycin in children as compared to adults in Poland [82]. 

Butenko et al. performed a study on *H. pylori*-infected children from Slovenia and identified the highest primary resistance rates to clarithromycin (23.4%) and metronidazole (20.2%), while only 2.8% of the strains were resistant to levofloxacin, 1% to amoxicillin, and none to tetracycline [83]. Additionally, 11.5% of the strains identified in this population were resistant to both clarithromycin and metronidazole, 2.9% to metronidazole and levofloxacin, and 2.8% clarithromycin and levofloxacin [83]. 

In German children, the most frequent resistance rates were also noticed for metronidazole (59%) and clarithromycin (45%), but interestingly, the authors also noticed a relatively high resistance rate to amoxicillin (20%) [84]. Furthermore, the study underlined multiple resistant strains in 16% of the cases [84]. 

On the contrary, in treatment-naïve *H. pylori*-infected children from Croatia, the greatest resistance rate was found for azithromycin (17.9%), followed by similar resistance rates to clarithromycin (11.9%) and metronidazole (10.1%), while the lowest rate was detected for amoxicillin (0.6%) [85] (Table 1). In Sweden, Jansson and Agardh also concluded that clarithromycin should not be used as first-line empirical therapy in children with detected *H. pylori* infection [86]. The same recommendation was stated by Manfredi et al. for Italian children, who also concluded that clarithromycin should be used only in children harboring *H. pylori* strains with proved clarithromycin susceptibility [87]. Moreover, a study from Portugal suggested that triple therapy consisting of amoxicillin and metronidazole as well as bismuth-based therapy might be the most suitable in eradicating pediatric *H. pylori* infection [88] (Table 1).

**Table 1 children-10-00403-t001:** *H. pylori* antimicrobial resistance among continents.

Continents	Countries	Antibiotics Resistance Rate (%)														
		Clarithromycin		Metronidazole		Amoxicillin		Levofloxacin		Clarithromycin + Metronidazole		Tetracycline		Others	Observations	
Asia	General (Savoldi et al. [33]	Primary 10%	Secondary 15%	Primary 51%	Secondary 44%	Primary 2% or less	Secondary –	Primary 2% or less	Secondary 24%	Primary –	Secondary –	Primary –	Secondary –	–	–	
	Turkey	24.8% to 36.7% [45] 40% [46]		33.7% to 35.5% [45]		0.9% [45]		23.7% [45]		–		3.5% [45]		–	Eradication rate 60% or less—standard triple therapy [43] Sequential therapy eradication rate 80% [46] 96.4% for bismuth-containing quadruple therapy, for levofloxacin-containing therapies → 95% [46]	
	China	55.2% [53]		71.3% [53]		-		18.4% [53]		*-*		*-*		60.9% for Rifampicin [53]	Eradication rates of the genotype-guided therapy were superior to that of phenotype-guided therapy—92% [52]	
		20.6% [54]		68.8% [54]		0 [54]		9% [54]		-		-		No resistance to furazolidone and gentamycin [54]	–	
	Iran	12.2% [66]		71% [66]		20.4% [66]		-		-		-		28.6% rifampicin 21.4% ampicillin 19% azithromycin 16.2% ciprofloxacin 15.3% erythromycin 8.4% for both tetracycline and furazolidone No resistance for nitrofurantoin [66]	Ciprofloxacin an option in eradicating multidrug-resistant *H. pylori* strains [65]	
	Korea	-		-		-		-		-		-		-	14-day quadruple therapy (bismuth, amoxicillin or tetracycline, metronidazole, and proton pump inhibitor—PPI) the optimal choice taking into account the lack of initial testing for *H. pylori* antibiotic susceptibility [58]; rifabutin an option in this population [58]	
	Vietnam	Primary 34.1% [59]	Secondary 74.9% [59]	Primary 69.4% [59]	Secondary 61.5% [59]	Primary 15% [59]	Secondary (9.5%) [59]	Primary 27.9% [59]	Secondary 45.7%	-		Primary 17.9% [59]	Secondary 23.5% [59]	-	Primary 48.8% for multidrug resistance [59]	Secondary 62.3% multidrug resistance [59]
		80.6% [60] 92.1% [61]		80.6% [60] 14.5% [61]		71.7% [60] 50% [61]		45.1% [60] 31.6% [61]		-		11.4% [60] 0% [61]		-	-	
	Israel	Primary 9.5% [62]	Secondary 29% [62]	Primary 32.6% [62]	Secondary 61% [62]	-		-		Primary 4.2% [62]	Secondary 18% [62]	-		-	-	
America (North, South, and Latin)	General (Savoldi et al. [33]	Primary 10%	Secondary 18%	-	-	Primary 10% [33]	Secondary 7% [33]	Primary 15%	Secondary 22% [33]	Primary 0 [33]	Secondary 0 [33]	-		–	–	
	Peru	79.6% [67]		-		-		-		-		-		-	-	
	Brasilia	19.5% [28]		40% [28]		10.4% [28]		-		-		0 for tetracycline [28]		0% for furazolidone [28]	-	
	Columbia	(8%) [69]		-		-		-		-		-		-		
	Chile	-		-		-		-		-		-		-	67% of the patients in whom the clarithromycin-based standard triple therapy failed carried A2143G mutations [70]	
Africa	General (Savoldi et al. [33]	15% [33]		91% [33]		38% [33]		14% [33]		-		13% [33]		-	-	
	Jaka et al. [71]	29.2% [71]		75.8% [71]		72.6% [71]		-		-		48.7% [71]		-	17.4% of the *H. pylori* isolated → resistant to quinolones [71]	
	Algeria	13% [72]		37% [72]		no resistance [72]		-		no resistance [72]		-		-	-	
	Kenya	-		-		-		-		-		-		-	Sequential therapy—higher eradication rate as compared to conventional triple therapy [73]	
Europe	General (Savoldi et al. [33]	Primary 18% [33]	Secondary 48% [33]	Primary 32% [33]	Secondary 48% [33]	-		Primary 11% [33]	Secondary 19% [33]	Primary 1% [33]	Secondary 18% [33]	-		-	-	
	Spain	44.9% [75] 34.7%		16.3% [75] 16.9% [76]		2% [75]		7.9% [75]		-		-		-	-	
	Poland	33.3% [77] 38.9% [78] 54.5% [82]		44.8% [77] 27.8% [78] 31.8%, [82]		-		-		-		-		1.9% simultaneous resistance to clarithromycin, metronidazole, and levofloxacin [77]	-	
	Slovenia	23.4% [83]		20.2% [83]		1% [83]		2.8% [83]		11.5% [83]		No resistance [83]		2.9% to metronidazole and levofloxacin and 2.8% clarithromycin and levofloxacin [83]	-	
	Germany	45% [84]		59% [84]		20% [84]		-		-		-		Multiple resistant strains in 16% of the cases [84]	-	
	Croatia	11.9% [85]		10.1% [85]		0.6% [85]		-		-		-		Azithromycin (17.9%) [85]	-	
	Sweden	-		-		-		-		-		-		-	Clarithromycin should not be used as first-line empirical therapy in children detected with *H. pylori* infection [86]	
	Italia	-		-		-		-		-		-		-	Clarithromycin should be used in children harboring *H. pylori* strains with proved clarithromycin susceptibility [87]	
	Portugal	-		-		-		-		-		-		-	Triple therapy consisting of amoxicillin and metronidazole as well as bismuth-based therapy might be the most suitable [88]	

In spite of the minor discrepancies between geographical areas of Europe and irrespective of the region, almost all studies stated that clarithromycin should be avoided in the eradication therapy of *H. pylori* in European children. Given the increasing trend for clarithromycin-resistant strains in all European countries, this antibiotic should be used only in those strains that were proven to be susceptible to this antibiotic. Similar to other continents, the high resistance rates of *H. pylori* to both clarithromycin and metronidazole might be explained by their use in a wide spectrum of medical conditions but most likely also by the mechanisms that *H. pylori* acquires continuously for escaping their mechanisms of action. 

## 7. *H. pylori* Resistance and Treatment Failure

Clarithromycin is used as a first option in the eradication of *H. pylori* infection, combined with either amoxicillin or metronidazole [89]. The most important mechanisms involved in *H. pylori* resistance to clarithromycin imply three main mutations in the 23S-rRNA gene: A2143G being the most frequent as well as A2142C and A2142 G [90]. Other mutations were also shown to contribute to this resistance, such as A2115G, A2144T, G2141A, G2223A, T2117C, T2182C, T2288C, and T2711C [38]. Additionally, infB and rpl22 genes were also proven to confer resistance to *H. pylori*, also expressing a synergistic effect when combined with 23S point mutations [91]. As we already mentioned in the introduction, the efflux system seems to be a major intrinsic contributor to clarithromycin resistance [38].

Amoxicillin is another first-line component of the standard triple therapy used for eradicating *H. pylori* infection worldwide. Although most of the bacteria express β-lactamases involved in inducing amoxicillin resistance, *H. pylori* differs from this category of bacteria since the β-lactamase activity is rarely detected in amoxicillin-resistant strains [38,92]. In terms of *H. pylori* amoxicillin resistance, point mutations to the penicillin-binding proteins were found to be involved in the underlying mechanism [38].

Metronidazole, a 5-nitroimidazole, is widely used for parasitic and bacterial infections, but it is also important for *H. pylori* eradication. Taking into account its complex mechanism of action, several mechanisms were hypothesized for explaining metronidazole resistance: (1) augmented activity of oxygen scavengers, (2) overactivity of the bacterial enzymes, (3) decreased activity of nitroreductases, as well as (4) diminished uptake of metronidazole [38].

Tetracyclines are used for *H. pylori* eradication as second- or third-line regimens when clarithromycin, amoxicillin, or metronidazole fail to be effective. Based on their limited use, tetracycline-resistant *H. pylori* strains are less commonly seen. Most frequently, *H. pylori* resistance to tetracyclines occurs as a result of mutations in the 16S rRNA gene at position 926–928 consisting of triple-base changes, i.e., AGA to TTC [38].

Levofloxacin is a potent antibiotic commonly used as a rescue therapy for *H. pylori* eradication. Regarding levofloxacin-resistant *H. pylori* strains, they usually express point mutations in gyrA gene at positions 87, 88, and 91 [93,94,95,96,97,98]. 

Based on all the above-mentioned facts along with other factors involved in *H. pylori* resistance, we might state that treatment failure is correlated with a wide range of complex factors interacting with each other, thus enabling *H. pylori* to persist within the gastric mucosa. 

## 8. Eradication Strategies for Preventing *H. pylori* Antimicrobial Resistance

Based on these findings, we consider it is highly important to implement in each country a continued surveillance of the prevalence of *H. pylori* antimicrobial resistance and its particularities in order to choose the most effective first-line eradication regimen tailored to the need of that particular population. Thus, metronidazole-based triple therapy might be useful as a first-line eradication regiment in areas with increased rates of clarithromycin resistance. Moreover, where available, bismuth quadruple therapy should be an option in areas with increased dual clarithromycin and metronidazole resistance [99]. Several studies suggested the replacement of proton pump inhibitors with potassium-competitive potent acid blocker vonoprazan as an alternative for increasing eradication rates in geographic areas experiencing high antibiotic resistance [100,101,102,103]. 

Additionally, in order to lower the rate of resistant infections, national healthcare authorities should also implement a close monitoring of antibiotics prescription for sustaining judicious antibiotics use. 

Tailored eradication therapy based on antibiotics susceptibility testing definitely represents the most effective therapeutic strategy for lowering *H. pylori* antimicrobial-resistance rates, but unfortunately, it carries great costs, and it is currently used mainly in studies or selected cases when available [99]. 

## 9. Conclusions

Although the current pediatric *H. pylori* prevalence has shown a decreasing trend worldwide, *H. pylori* antimicrobial-resistance rates have been noticed to have a persistent increase. In terms of Asia, metronidazole and clarithromycin show the greatest resistance rates, but there is also a surprisingly high resistance rate for levofloxacin. Similar high resistance rates for clarithromycin, metronidazole, and levofloxacin were also noticed in America, and it was also proven that, if used together, clarithromycin and metronidazole might still be effective in eradicating *H. pylori* infection in American children. The findings regarding increased clarithromycin- and metronidazole-resistance rates were also confirmed for the African pediatric population but not for levofloxacin, whose resistance rates were proven to be lower as compared to the other two continents. Moreover, reports from Africa indicated that sequential therapy as well as the combination between amoxicillin and high doses of metronidazole might be useful in effective *H. pylori* eradication. In European children infected with *H. pylori*, the resistance rates were also noticed to be higher for clarithromycin and metronidazole when compared to levofloxacin, amoxicillin, or tetracycline. It is rather alarming that the reports from these areas suggest that clarithromycin resistance rates are even higher in *H. pylori*-positive children as compared to infected adults. Still, the combination between clarithromycin and amoxicillin remains a viable option for treating *H. pylori* infection in children from Europe harboring susceptible strains along with sequential therapy, bismuth-based therapy, or even amoxicillin combined with metronidazole for those infected with strains resistant to clarithromycin. Even more concerning is the increasing emergence of multidrug-resistant *H. pylori* strains in pediatric patients worldwide. 

Aside from *H. pylori* resistance mechanisms, the differences in antibiotic use among continents and countries worldwide is also responsible for the discrepancies in terms of antimicrobial-resistance patterns. Thus, global judicious antibiotic use might decrease *H. pylori* antimicrobial resistance. Taking into account the high costs and the invasiveness related to the antimicrobial-resistance studies, which cannot be performed in daily practice, empirical eradication treatment remains the most commonly used option in pediatric patients. Therefore, we emphasize the need for further studies on larger pediatric populations living in different geographical areas in order to identify the peculiarities regarding antimicrobial-resistance patterns in these populations for increasing the efficacy of empirical eradication strategies in terms of *H. pylori* infection. 

## Data Availability

Not applicable.

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
