# Peer review of "Current Worldwide Trends in Pediatric *Helicobacter pylori* Antimicrobial Resistance"

_children, 2023, doi:10.3390/children10020403_

Round 1
Reviewer 1 Report
A nice review regarding pediatric infection of H. pylori. However, I have several comments before this manuscript can be accepted.
Introduction section is too long and sometimes not relevant to this review. For example, line 35-74 can be simplified and some lines can be removed.
Line 75 to 79 is not necessary and can be deleted.
Line 81 to 86 can be further simplified.
What is the relevance of Line 87 to 94? It describes microbiota diversity and I think can be simplified or removed entirely to make writing more concise on antibiotic resistance.
While Table 1 is helpful to show the trend of antibiotic resistance worldwide among children, it will be nicer if the authors can draw graph that shows how the trend of antibiotic resistance among children worldwide has changed in recent years.
Author Response
The 30th January 2023
To Editor/Reviewers of Children,
Dear Editor/Reviewers,
Please find attached a revised version of the manuscript entitled: “Current worldwide trends in pediatric Helicobacter pylori antimicrobial resistance” written by Reka Borka Balas, Lorena Elena Meliț and Cristina Oana Mărginean, Manuscript ID: children - 216708.
Firstly, we thank very much the reviewers for their valuable comments and suggestions in order to improve our paper.
Following the reviewers’ concerns and observations, we made some modifications to the initial version of our manuscript, which we described in great detail, according to their recommendations, highlighting them in blue in the attached manuscript as it follows:
Reviewer 1
Comment 1
A nice review regarding pediatric infection of H. pylori. However, I have several comments before this manuscript can be accepted.
Answer 1
Thank you very much for your positive comments valuable time spent on assessing our manuscript.
Comment 2
Introduction section is too long and sometimes not relevant to this review. For example, line 35-74 can be simplified and some lines can be removed.
Answer 2
We simplified the lines 35-74 according to your recommendations and we also deleted the unnecessary information.
Comment 3
Line 75 to 79 is not necessary and can be deleted.
Answer 3
We deleted the lines 75-79 according to your recommendations.
Comment 4
Line 81 to 86 can be further simplified.
Answer 4
We simplified lines 81-86.
Comment 5
What is the relevance of Line 87 to 94? It describes microbiota diversity and I think can be simplified or removed entirely to make writing more concise on antibiotic resistance.
Answer 5
We removed lines 87-94 following your recommendations.
Comment 6
While Table 1 is helpful to show the trend of antibiotic resistance worldwide among children, it will be nicer if the authors can draw graph that shows how the trend of antibiotic resistance among children worldwide has changed in recent years.
Answer 6
We completely agree with your suggestion, but we chose to show the trends in the table due to the fact that we did not find clear evidence all around the world and not for each antibiotic and thus, it is impossible to draw a graph. Therefore, please kindly agree to the table.
We must also mention that our manuscript was also revised for language errors.
Respectfully,
Lecturer Lorena Elena Meliț, MD, PhD
Reviewer 2 Report
The review article – “Current worldwide trends in pediatric Helicobacter pylori antimicrobial resistance”- by Reka Borka Balas et al aims to provide current patterns of antimicrobial resistance of H.pylori in children.
However, the article Name, provided Aim (lacks pediatric/children population) and Manuscript section does not correlate.
The abstract needs to be rewritten. It is noninformative, the first sentence regarding mechanisms does not reply to the information on resistance rates worldwide, which we do not have in the abstract, only the highest/lowest etc.
Lines 10-11: Helicobacter pylori (H. pylori) is a tricky bacterium which learnt to develop several antimicrobial resistance mechanisms. – Please provide better and more professional sentences.
The Introduction contains general information and conclusions without specific elucidations (clever pathogen? Why is more clever than any other?). There is no specific information regarding the proposed Name and Aim of the Manuscript.
Introduction: Helicobacter pylori (H. pylori) usually infects children – when, how, and why?
The aim of this review is to assess the patterns of H. pylori antimicrobial resistance patterns worldwide. – the suggested aim is not correlating with the manuscript.
There is no explanation of how the authors organised the review, searched the literature, performed processing of articles and information, detected mechanisms/patterns, etc.
The authors stated that they would assess antimicrobial patterns, without defining whether they are phenotypic or genotypic. In the sections of geographic regions, we have a general listing of resistance rate information without specific discussion on methods, patients, therapy, and acclaimed patterns (molecular or phenotypic).
The conclusion is also a generalisation which could be used for any bacteria in medical microbiology.
As the topic of the suggested manuscript is important and interesting, I am suggesting that the authors rewrite the article according to the PRISMA (Preferred Reporting Items for Systematic Reviews and Meta-Analyses) guidelines - Moher D, Liberati A, Tetzlaff J, Altman DG, PRISMA Group Preferred reporting items for systematic reviews and meta-analyses: the PRISMA statement. BMJ. 2009;339:b2535.
Author Response
The 30th January 2023
To Editor/Reviewers of Children,
Dear Editor/Reviewers,
Please find attached a revised version of the manuscript entitled: “Current worldwide trends in pediatric Helicobacter pylori antimicrobial resistance” written by Reka Borka Balas, Lorena Elena Meliț and Cristina Oana Mărginean, Manuscript ID: children - 2167081.
Firstly, we thank very much the reviewers for their valuable comments and suggestions in order to improve our paper.
Following the reviewers’ concerns and observations, we made some modifications to the initial version of our manuscript, which we described in great detail, according to their recommendations, highlighting them in blue in the attached manuscript as it follows:
Reviewer 2
Comment 1
The review article – “Current worldwide trends in pediatric Helicobacter pylori antimicrobial resistance”- by Reka Borka Balas et al aims to provide current patterns of antimicrobial resistance of H.pylori in children.
However, the article Name, provided Aim (lacks pediatric/children population) and Manuscript section does not correlate.
Answer 2
We apologize for our mistake. We added the word ‘pediatric’ in the aim and each section title.
Comment 3
The abstract needs to be rewritten. It is noninformative, the first sentence regarding mechanisms does not reply to the information on resistance rates worldwide, which we do not have in the abstract, only the highest/lowest etc.
Answer 3
Comment 4
Lines 10-11: Helicobacter pylori (H. pylori) is a tricky bacterium which learnt to develop several antimicrobial resistance mechanisms. – Please provide better and more professional sentences.
Answer 4
We rephrased completely the abstract according to your recommendations:
‘Helicobacter pylori (H. pylori) has acquired several resistance mechanisms in order to escape the currently used eradication regimens such as mutations that impair the replication, recombination and transcription of DNA, the antibiotics capability to interact with protein synthesis and ribosomal activity, the adequate redox-state of bacterial cells, or the penicillin binding proteins. The aim of this review was to identify the differences in pediatric H. pylori antimicrobial resistance trends between continents and countries of the same continent. In Asian pediatric patients, the greatest antimicrobial resistance was found to metronidazole (>50%), probably due to its wide use for parasitic infections. In Turkey the resistance rates were found to be similar for both metronidazole and clarithromycin (30%). The reports from China underlined increased resistance rates for metronidazole (71.3%), rifampicin (60.9%), and clarithromycin (55.2%), but no resistant strains to amoxicillin, tetracycline or furazolidone. Studies in Iranian children recommended ciprofloxacin as an alternative for H. pylori eradication, while those in Korea highlighted that bismuth-based quadruple therapy might be an optimal choice. The findings from Vietnam remain contradictory since certain authors proved that metronidazole has the highest resistance rate (69.4%), while others revealed high resistance to clarithromycin and amoxicillin. In both Iranian and Israeli children, H. pylori resistance rate was the highest for metronidazole (up to 61%), but Iranian studies reported the lowest resistance rate to clarithromycin (12.2%). The scarce evidence for America revealed that H. pylori strains display an increased resistance to clarithromycin (up to 79.6%), but not all studies agreed on this statement. Pediatric patients from Africa also presented the greatest resistance rate to metronidazole (91%), but the results in terms of amoxicillin remain contradictory. Nevertheless, the lowest resistance rates in most of the African studies was found for quinolones. Among European children, the most frequent antimicrobial resistance was also noticed for metronidazole and clarithromycin (up to 59% and 45%), but with a predominance for clarithromycin as compared to other continents’.
Comment 5
The Introduction contains general information and conclusions without specific elucidations (clever pathogen? Why is more clever than any other?). There is no specific information regarding the proposed Name and Aim of the Manuscript.
Answer 5
We removed the unnecessary paragraphs from the introduction.
Comment 6
Introduction: Helicobacter pylori (H. pylori) usually infects children – when, how, and why?
Answer 6
We apologize for the lack of clarity. We meant that this infection occurs during childhood and its prevalence increases with age. Thus, we rephrased: ‘Helicobacter pylori (H. pylori) infection occurs usually during early childhood and its prevalence increases with age being probably the most studied infection worldwide’.
Comment 7
The aim of this review is to assess the patterns of H. pylori antimicrobial resistance patterns worldwide. – the suggested aim is not correlating with the manuscript.
Answer 7
Thank you for your valuable suggestion. We rephrased the aim as it follows: ‘The aim of this review is to assess the trends of pediatric H. pylori antimicrobial resistance worldwide’.
Comment 8
There is no explanation of how the authors organised the review, searched the literature, performed processing of articles and information, detected mechanisms/patterns, etc.
Answer 8
In order to clarify the details you mentioned above, we introduced the following section in the revised form of our manuscript:
‘2. Literature search
We performed an electronic search of the literature on the lines of search for a narrative overview based on three databases: PubMed, EMBASE and Google Scholar, using the following search terms: ‘Helicobacter pylori antimicrobial resistance’, ‘children’, as well as names of countries from each continent. We included articles published in the last ten years involving pediatric patients on the topic of H. pylori antimicrobial resistance in order to underline the differences in resistance trends worldwide. We critically assessed all included articles depending on the suitability of the methods used to check the hypothesis, the key results, the interpretation of the results, the quality of the results, the limitations of the study, and the relevance of the conclusions to the field[35]. We used the critical reading tools for scientific article proposed by Jean-Baptist du Prel[36] et al involving the assessment of the design and structure of each article, the relevance of each section, but also the possible sources of limitation and bias. Based on these theories, the most reliable critical reading tools used to assess the articles included in this review were analysis and inference. All articles that did not match these criteria were excluded in the study, as well as those written in other languages than English or which has only the abstract available. When assessing the articles, we also identified supplementary references that matched our criteria and we performed a manual search in the reference list of each articles retrieved in the initial round of search. For these articles, we followed the same method as described above before including them in the review. Eventually, we synthetized the results from all articles included in the study in order to underlined both the similarities and discrepancies between them for providing an objective narrative overview regarding the trends of H. pylori antimicrobial resistance worldwide.’
Comment 9
The authors stated that they would assess antimicrobial patterns, without defining whether they are phenotypic or genotypic. In the sections of geographic regions, we have a general listing of resistance rate information without specific discussion on methods, patients, therapy, and acclaimed patterns (molecular or phenotypic).
Answer 9
We apologize for our confusing statement. We restated the aim of our manuscript avoiding the word pattern: ‘The aim of this review is to assess the trends of pediatric H. pylori antimicrobial resistance worldwide.’. We apologize once more for not stating clearly what was the aim of our study, but this objective narrative review was not meant to assess the methods, therapies and molecular or phenotypic patterns.
Comment 10
The conclusion is also a generalisation which could be used for any bacteria in medical microbiology.
Answer 10
We rephrased the conclusions according to your recommendations: ‘Although the current trends of pediatric H. pylori prevalence showed a decreasing trend worldwide, H. pylori antimicrobial resistance rates were noticed to have a persistent increase. In terms of Asia, metronidazole and clarithromycin own the greatest resistance rates, but with a surprisingly high resistance rate also for levofloxacin. Similar high resistance rates for clarithromycin, metronidazole and levofloxacin were also noticed in America, being also proved that if used together clarithromycin and metronidazole might still be effective in eradicating H. pylori infection in American children. The findings regarding increased clarithromycin and metronidazole resistance rates were also confirmed for African pediatric population, but not for levofloxacin whose resistance rates were proved to be lower as compared to the other two continents. Moreover, reports from African indicated that sequential therapy, as well as the combination between amoxicillin and high doses of metronidazole might be useful in H. pylori effective eradication. In European children infected with H. pylori, the resistance rates were also noticed to be higher for clarithromycin and metronidazole when compared to levofloxacin, amoxicillin or tetracycline. It is rather alarming that the reports from these areas suggested that clarithromycin resistance rates are even higher in H. pylori positive children as compared to infected adults. Still, the combination between clarithromycin and amoxicillin remains a viable option for treating H. pylori infection in children from Europe harboring susceptible strains along with sequential therapy, bismuth-based therapy or even amoxicillin combined with metronidazole for those infected with resistant strains to clarithromycin. Even more concerning is the increasing emergence of multidrug resistance H. pylori strains in pediatric patients worldwide.’
Comment 11
As the topic of the suggested manuscript is important and interesting, I am suggesting that the authors rewrite the article according to the PRISMA (Preferred Reporting Items for Systematic Reviews and Meta-Analyses) guidelines - Moher D, Liberati A, Tetzlaff J, Altman DG, PRISMA Group Preferred reporting items for systematic reviews and meta-analyses: the PRISMA statement. BMJ. 2009;339:b2535.
Answer 11
Thank you for your valuable suggestion, but this meant to be a narrative review, and thus our intention was to increase awareness regarding the judicious use of eradication therapies. Therefore, please accept this review as a narrative review and if possible we kindly ask you to acknowledge its importance in its current form since we intend in the future to design a more accurate systematic review of all studies published in the literature regarding the mechanism of H. pylori antimicrobial resistance. Based on the fact that the aim of this review was to characterize only the differences regarding antimicrobial trends among different countries, we consider it is more valuable in this form since it comprised unsystematized studies on children infected with H. pylori worldwide. In order to perform a PRISMA review, we would have to remove several studies from this analysis and it would decrease its importance for increasing the awareness regarding this topic. Nevertheless, we definitely appreciate your suggestion, and we will take into consideration for our next work. Once more, we kindly ask you to accept our narrative review in its current form.
We must also mention that our manuscript was also revised for language errors.
Respectfully,
Lecturer Lorena Elena Meliț, MD, PhD
Reviewer 3 Report
Dear authors
Please, find the attached PDF file.

Author Response
The 30th January 2023
To Editor/Reviewers of Children,
Dear Editor/Reviewers,
Please find attached a revised version of the manuscript entitled: “Current worldwide trends in pediatric Helicobacter pylori antimicrobial resistance” written by Reka Borka Balas, Lorena Elena Meliț and Cristina Oana Mărginean, Manuscript ID: children - 2167081.
Firstly, we thank very much the reviewers for their valuable comments and suggestions in order to improve our paper.
Following the reviewers’ concerns and observations, we made some modifications to the initial version of our manuscript, which we described in great detail, according to their recommendations, highlighting them in blue in the attached manuscript as it follows:
Reviewer 3
Comment 1
The authors must mention the resistance oh H. pylori to antibiotics in a serial manner of resistance evolution instead of mentioning the resistance in each country.
Answer 1
Thank you very much for all your efforts spent on assessing our manuscript. We must mention from the beginning that the aim of this review was not to study the resistance mechanisms, but to assess the differences in terms of H. pylori antimicrobial resistance trends in different geographical areas. Thus, we rephrased the aim of this study for avoiding confusions: ‘The aim of this review is to assess the trends of pediatric H. pylori antimicrobial resistance worldwide.’ Nevertheless, we added several information regarding H. pylori antimicrobial resistance mechanisms according to your suggestions below.
Comment 2
Moreover, the abstract should include items other than the incidence of resistance in different countries.
Answer 2
As we already stated this was the aim of this narrative review, to assess the differences between resistance incidence to different antibiotics commonly used for the eradication of this bacterium, and not to study the mechanisms involved in this antimicrobial resistance. Nevertheless, certain details regarding the mechanisms were described in the abstract, which was rephrased:
‘Helicobacter pylori (H. pylori) has acquired several resistance mechanisms in order to escape the currently used eradication regimens such as mutations that impair the replication, recombination and transcription of DNA, the antibiotics capability to interact with protein synthesis and ribosomal activity, the adequate redox-state of bacterial cells, or the penicillin binding proteins. The aim of this review was to identify the differences in pediatric H. pylori antimicrobial resistance trends between continents and countries of the same continent. In Asian pediatric patients, the greatest antimicrobial resistance was found to metronidazole (>50%), probably due to its wide use for parasitic infections. In Turkey the resistance rates were found to be similar for both metronidazole and clarithromycin (30%). The reports from China underlined increased resistance rates for metronidazole (71.3%), rifampicin (60.9%), and clarithromycin (55.2%), but no resistant strains to amoxicillin, tetracycline or furazolidone. Studies in Iranian children recommended ciprofloxacin as an alternative for H. pylori eradication, while those in Korea highlighted that bismuth-based quadruple therapy might be an optimal choice. The findings from Vietnam remain contradictory since certain authors proved that metronidazole has the highest resistance rate (69.4%), while others revealed high resistance to clarithromycin and amoxicillin. In both Iranian and Israeli children, H. pylori resistance rate was the highest for metronidazole (up to 61%), but Iranian studies reported the lowest resistance rate to clarithromycin (12.2%). The scarce evidence for America revealed that H. pylori strains display an increased resistance to clarithromycin (up to 79.6%), but not all studies agreed on this statement. Pediatric patients from Africa also presented the greatest resistance rate to metronidazole (91%), but the results in terms of amoxicillin remain contradictory. Nevertheless, the lowest resistance rates in most of the African studies was found for quinolones. Among European children, the most frequent antimicrobial resistance was also noticed for metronidazole and clarithromycin (up to 59% and 45%), but with a predominance for clarithromycin as compared to other continents.’
Comment 3
Talk about the pathogenesis and the virulence factors role in this pathogenesis.
Answer 3
Thank you for your suggestion. We mentioned several bacterial particularities involved in H. pylori antimicrobial resistance: ‘Certain bacterial peculiarities were also revealed to reduce susceptibility to antibiotics like its ability to shift between rod-shaped active bacteria and dormant resting coccoidal state as a response to antibiotics, but also the capacity to form biofilms on gastric mucosa surface after its penetration[34].’
Comment 4
Talk about resistance mechanisms
Answer 4
Thank you for your suggestion. We introduced a paragraph in the discussions section about resistance mechanisms: ‘Several mechanisms were found to be involved in H. pylori antimicrobial resistance, which in fact are common to other bacterial infections in humans, such as mutations that impair the replication, recombination and transcription of DNA; mutations that affect the antibiotics capability to interact with protein synthesis and ribosomal activity; mutations altering the adequate redox-state of bacterial cells resulting in impaired activity of oxireductases; as well as mutations that change penicillin binding proteins contributing to peptidoglycan biosynthesis and specific target of β-lactams action[32]. Moreover, outer membrane poring and efflux systems were also proved to strongly contribute to drugs tolerance due to their ability maintain a low concentration of toxic agent within the bacterial cells resulting in a decreased capacity of antibiotics to kill bacteria[33]. Certain bacterial peculiarities were also revealed to reduce susceptibility to antibiotics like its ability to shift between rod-shaped active bacteria and dormant resting coccoidal state as a response to antibiotics, but also the capacity to form biofilms on gastric mucosa surface after its penetration[34].’
Comment 5
What will this research add to this crisis?
Answer 5
This research is meant to increase the awareness regarding the judicious use of antibiotics worldwide in order to design more effective eradication regimens based on the trends of H. pylori resistance peculiarities in a certain geographic area. Thus, this review would add great value not only to the scientific area, but also in clinical practice due to the synthetization and stratification of resistance trends in each geographic area. The clinicians working in this area might guide their therapeutic choice on the findings reported in this review in order to increase the eradication rate. Moreover, this research might be a starting point for further research aiming to elucidate the underlying mechanisms for these differences between continents and even countries of the same continent.
Comment 6
The authors should correlate the resistance with the treatment failure.
Answer 6
Although we mentioned and we strengthen once more that this review was not aimed to assess resistance mechanisms, we followed your valuable recommendation, and we introduced a short discussion about the correlation between resistance and treatment failure:
‘7. The correlation between H. pylori resistance and treatment failure
Clarithromycin is used as a first option in the eradication of H. pylori infection combined with either amoxicillin or metronidazole[83]. The most important mechanisms involved in H. pylori resistance to clarithromycin imply three main mutations in 23S-rRNA gene: A2143G, the most frequent, A2142C, and A2142 G[84]. Other mutations were also encountered to contribute to this resistance such as A2115G, A2144T, G2141A, G2223A, T2117C, T2182C, T2288C, and T2711C[32]. Additionally, infB and rpl22 genes were also proved to confer resistance to H. pylori expressing also a synergistic effect when combined with 23S point mutations[85]. As we already mentioned in the introduction, the efflux system seem to be a major intrinsic contributor to clarithromycin resistance[32].
Amoxicillin is another first line component of the standard triple therapy used for eradicating H. pylori infection worldwide. Although most of the bacteria express β-lactamases involved in inducing amoxicillin resistance, H. pylori differs from this category of bacteria since the β-lactamase activity was rarely detected in amoxicillin resistant strains[32,86]. In terms of H. pylori amoxicillin resistance, point mutations to the penicillin-binding proteins were found to be involved in the underlying mechanism[32].
Metronidazole, a 5-nitroimidazole is widely used for parasitic and bacterial infections, but it is also important for H. pylori eradication. Taking into account its complex mechanism of action, several mechanisms were hypothesized for explaining metronidazole resistance: 1) augmented activity of oxygen scavengers, 2) overactivity of the bacterial enzymes, 3) decreased activity of nitroreductases, as well as 4) diminished uptake of metronidazole[32].
Tetracyclines are used for H. pylori eradication as second or third-line regimens when clarithromycin, amoxicillin or metronidazole fail to be effective. Based on their limited use, H. pylori tetracycline resistant strains are less commonly seen. Most frequently, H. pylori resistance to tetracyclines occurs as a result of mutations in the 16S rRNA gene at position 926-928 consisting of triple-base changes, i.e. AGA to TTC[32].
Levofloxacin is a potent antibiotic commonly used as a rescue therapy for H. pylori eradication. Regarding H. pylori levofloxacin resistant strains they usually express point mutations in gyrA gene at positions 87, 88 and 91[87–92].
’
Comment 7
Under all titles, the authors must mention the reasons for resistance and its drawbacks in such areas targeting children.
Answer 7
Thank you for your suggestion. We introduced an opinion statement and the end of each section regarding potential reasons for the identified resistance trends.
Comment 8
I suggest adding new title about the control of such resistance via different ways.
Answer 8
Thank you for your suggestion. We introduced a section regarding the topic you mentioned above:
‘6. Eradication strategies for preventing H. pylori antimicrobial resistance
Based on these findings, we consider it is highly important to implement in each country a continued surveillance of the prevalence of H. pylori antimicrobial resistance and its particularities in order to choose the most effective firs-line eradication regimen tailored to the need of that particular population. Thus, metronidazole-based triple therapy might be useful as a first-line eradication regiment in areas with increased rates of clarithromycin resistance. Also, where available, bismuth quadruple therapy should be an option in areas with increased dual clarithromycin and metronidazole resistance[93]. Several studies pointed out the replacement of proton pump inhibitors with potassium-competitive potent acid blocker vonoprazan as an alternative for increasing eradication rates in geographic areas experiencing high antibiotic resistance[94–97].
Additionally, in order to lower the rate of resistant infections, national healthcare authorities should also implement a close monitoring of antibiotics prescription sustaining judicious antibiotics use.
Tailored eradication therapy based on antibiotics susceptibility testing definitely represents the most effective therapeutic strategy for lowering H. pylori antimicrobial resistance rate but unfortunately it carries great costs and it is currently used mainly in studies or selected cases when available[93].’
Comment 9
Why did the authors mention only these drugs only?
Answer 9
We chose the most commonly used drugs worldwide for the eradication of H. pylori infection in pediatric patients.
Comment 10
Add columns specifying the mechanism of action of resistance to such antibiotic.
Answer 10
Please accept that this was not the aim of our study and therefore for this review at least, we do not consider it is suitable to introduce such columns in the table.
Comment 11
This category must be specified all over the whole article as it is the subject of study.
Answer 11
We deeply appreciate your suggestions, but it was not the subject of this study and we apologize once more for confounding you asking you kindly to accept that this was a narrative review aimed to observe and synthetize the main similarities and differences regarding H. pylori resistance to different antibiotics used worldwide for its eradication. Still, we will take into account your advice for our future work and we intend to write an article regarding the mechanisms involved in H. pylori antimicrobial resistance. Thank you again!
We must also mention that our manuscript was also revised for language errors.
Respectfully,
Lecturer Lorena Elena Meliț, MD, PhD
Round 2
Reviewer 1 Report
No more comment from me.
Author Response
The 5th February 2023
To Editor/Reviewers of Children,
Dear Editor/Reviewers,
Please find attached a revised version of the manuscript entitled: “Current worldwide trends in pediatric Helicobacter pylori antimicrobial resistance” written by Reka Borka Balas, Lorena Elena Meliț and Cristina Oana Mărginean, Manuscript ID: children - 216708.
Firstly, we thank very much the reviewers for their valuable comments and suggestions in order to improve our paper.
Following the reviewers’ concerns and observations, we made some modifications to the initial version of our manuscript, which we described in great detail, according to their recommendations, highlighting them in blue in the attached manuscript as it follows:
Reviewer 1
Comment 1
No more comment from me.
Answer 1
Thank you very much for reviewing the revised version of our manuscript and for appreciating our work.
We must also mention that our manuscript was also revised for language errors.
Respectfully,
Lecturer Lorena Elena Meliț, MD, PhD
Reviewer 3 Report
Dear authors
Please, find the attached file

Author Response
The 5th February 2023
To Editor/Reviewers of Children,
Dear Editor/Reviewers,
Please find attached a revised version of the manuscript entitled: “Current worldwide trends in pediatric Helicobacter pylori antimicrobial resistance” written by Reka Borka Balas, Lorena Elena Meliț and Cristina Oana Mărginean, Manuscript ID: children - 216708.
Firstly, we thank very much the reviewers for their valuable comments and suggestions in order to improve our paper.
Following the reviewers’ concerns and observations, we made some modifications to the initial version of our manuscript, which we described in great detail, according to their recommendations, highlighting them in blue in the attached manuscript as it follows:
Reviewer 3
Comment 1
The authors must mention evolution of resistance of H. pylori.
Answer 1
Thank you very much for your suggestion, but we cannot introduce new details in the abstract since we are word limited and we have only 300 words for the abstract.
Comment 2
Are these resistances for pediatric H. pylori only?. Please, clarify. Moreover, the authors should add the mechanisms for this resistance.
Answer 2
Yes, these resistance rates are for pediatric patients, and according to your recommendation we introduced this in the abstract. As we already mentioned above we cannot expand the abstract. Moreover, our manuscript was not aimed to assess the mechanism for H. pylori resistance. Please be kind and accept this fact.
Comment 3
Talk about the pathogenesis and the virulence factors role in this pathogenesis.
Answer 3
Thank you for your suggestion. We further introduced a paragraph regarding pathogenesis and virulence factors: ‘A recent study underlined that pediatric patients harboring cagA and cagE positive H. pylori strains are less susceptible to H. pylori eradication treatment[14]. Similar findings were reported also by Karbalaei et al who emphasized that cagA positive strains express increased resistance to metronidazole, while those carrying vacA s1m1 genotype resulted in increased resistance to amoxicillin and levofloxacin[15]. Moreover, Karabiber et al pointed out that cagE and vacA s1 a/m2 genotypes represent major virulence factors associated with increased antibiotic resistance[16]. Aside from cagA, vacA s1m1 and cagE, other studies proved that iceA2 and babA2 are also associated with H. pylori-positive gastritis severity[17,18].’
Comment 4
What will this research add to this crisis?
Answer 4
We rephrased better our aim in order to underline the valuable information provided by this manuscript: ‘The aim of this review was to assess the trends of pediatric H. pylori antimicrobial resistance worldwide in order to guide the eradication treatment according to geographical peculiarities.’
Comment 5
This section is not vital for the study and didn't add conclusions to the study.
Answer 5
This section was requested by reviewer 2 in order to describe the material and methods for this review and thus please be kind and accept it.
Comment 6
The authors must mention the drawbacks of resistances in different countries targeting children.
Answer 6
At the end of each section we already introduced a paragraph regarding the drawbacks for explaining the different resistance rates in different countries:
‘The increased H. pylori resistance rates to metronidazole in Asia might be related to its wide use as an antiparasitic agent in these areas since it is well-known that parasitic infections are frequent in Asian population. Surprisingly, most of the studies in these countries reported relatively high resistance rates to levofloxacin which might result in reducing the successful of eradication regimens based on the fact that levofloxacin is considered a rescue therapy.’
‘Although in American children, most of the studies revealed H. pylori resistant strains to clarithromycin and metronidazole, several reported indicated the contrary imposing the need for further studies in order to delineate clear recommendations for eradicating H. pylori infection in different areas of America (Table 1). The reported high resistance rates to both metronidazole and clarithromycin, as well as the increasing trend regarding levofloxacin resistance rate in this geographic area might be related to their wide use for other bacterial infections probably sometimes unjustified. Nevertheless, studies indicated that if used together, clarithromycin and metronidazole seem to maintain their effectiveness in terms of H. pylori eradication.’;
‘Undoubtedly, similar to the reports from Asia, H. pylori resistance to metronidazole is a major problem in children from Africa most-likely due to its wide use in parasitic infections, which are highly prevalent in this population (Table 1). Nevertheless, as compared to Asian and American children, the resistance rates for levofloxacin in African pediatric population were found to be lower.’;
‘In spite of the minor discrepancies between geographical areas of Europe, almost all studies irrespectively of the region stated that clarithromycin should be avoided in the eradication therapy of H. pylori in European children. Given the increasing trend for clarithromycin resistance strains in all European countries, this antibiotic should be used only in those strains which were proven to be susceptible to this antibiotic. Similar to other continents, the high resistance rates of H. pylori to both clarithromycin and metronidazole might be explained by their use in a wide-spectrum of medical conditions, but most-likely also by the mechanisms that H. pylori acquires continuously for escaping their mechanisms of action.’.
Comment 8
Add columns specifying the mechanism of action of resistance to such antibiotic.
Answer 8
Please accept that this was not the aim of our study and therefore for this review at least, we do not consider it is suitable to introduce such columns in the table. Nevertheless, following your thorough recommendations, we intend to design another review article only for assessing the resistance mechanisms.
Comment 9
Why did the authors mention only these drugs only?
Answer 9
We chose to mention only these drugs since they are the most important and the most commonly used worldwide for the eradication of H. pylori infection in pediatric patients.
Comment 11
Clarify this correlation instead of mentioning the antibiotics and the mechanisms underlying their resistances.
Answer 11
We deeply appreciate your suggestions, but we mentioned the underlying resistance mechanisms for each antibiotic exactly for explaining the correlation between H. pylori resistance. Therefore, each mutations are meant to explain the mechanisms. Nevertheless, we removed the world correlation from the section title and we added a paragraph at the end of this section: ‘Based on all the above-mentioned facts along with other factors involved in H. pylori resistance, we might state that treatment failure is correlated with a wide-range of complex factors interacting with each other for enabling H. pylori to persist within the gastric mucosa.’
We must also mention that our manuscript was also revised for language errors.
Respectfully,
Lecturer Lorena Elena Meliț, MD, PhD
